# Augmenting Advanced Analytics into Enterprise Systems: A Focus on Post-Implementation Activities

**Ahmed Elragal** and **Hossam El-Din Hassanien** *

Department of Computer Science, Electrical and Space Engineering (SRT), Luleå University of Technology,
SE-971 87 Luleå, Sweden; ahmed.elragal@ltu.se
* Correspondence: hossam.hassanien@ltu.se

**Abstract:** An analytics-empowered enterprise system looks to many organizations to be a far-fetched target, owing to the vast amounts of factors that need to be controlled across the implementation lifecycle activities, especially during usage and maintenance phases. On the other hand, advanced analytics techniques such as machine learning and data mining have been strongly present in academic as well as industrial arenas through robust classification and prediction. Correspondingly, this paper is set out to address a methodological approach that works on tackling post-live implementation activities, focusing on employing advanced analytics techniques to detect (business process) problems, find and recommend a solution to them, and confirm the solution. The objective is to make enterprise systems self-moderated by reducing the reliance on vendor support. The paper will profile an advanced analytics engine architecture fitted on top of an enterprise system to demonstrate the approach. Employing an advanced analytics engine has the potential to support post-implementation activities. Our research is innovative in two ways: (1) it enables enterprise systems to become self-moderated and increase their availability; and (2) the IT artifact i.e., the analytics engine, has the potential to solve other problems and be used by other systems, e.g., HRIS. This paper is beneficial to businesses implementing enterprise systems. It highlights how enterprise systems could be safeguarded from retirement caused by post-implementation problems.

**Keywords:** enterprise systems; advanced analytics; post-live activities; post-implementation

## 1. Introduction

An enterprise resource planning (ERP) system is a type of enterprise system (ES) that acts as an integrated set of programs to provide support for core organizational functions such as manufacturing and logistics, finance and accounting, sales and marketing, and human resources. An ERP system helps the different parts of the organization to share data, reduce operating costs, and improve the efficiency of business processes [1,2]. Despite these benefits, many ERP systems fail, owing to implementation difficulties [3,4]. It has also been observed from the literature that ERP post-implementation has not received much research focus [1], which is a point that had been made by early [5,6] and recent [7] research alike. The latter statement defines the basis of our research work as well as being the triggering point towards the awareness of the problem at hand, for which our focus is on post-implementation failures.

On the other hand, the requirement to transform liabilities of data into actionable enterprise decisions has provided a fertile foundation for various transformational offspring analytical technologies as a direct consequence of big data analytics (BDA) [8]. With the rise of analytical technologies and the ability to execute complex algorithms on huge amounts data; algorithmic accuracy has become quite important. Correspondingly, considerable improvements have been experienced through typical evolution on analytical algorithms. The incidence of algorithmic evolution can be seen clearly in the

case of artificial neural networks (ANNs) evolving into deep-learning algorithms [9]. Deep learning enables hierarchal methods to extract complex features from simple abstractions.

Employing advanced analytics algorithms together with enabling components within typical BDA architectures has the potential to support post-live ERP systems. Correspondingly, this would command the development of artificially intelligent agents that work on analyzing ERP system data to prevent process failure. That said, this marks the first academic attempt to build artificially intelligent agents that empower ESs for post-implementation activities.

The following sections will introduce the concepts as well as discuss potential solutions and research methodologies to be adopted in this research activity through the methodology of design-science research. The paper is therefore structured as follows: Section 2 will shed light on ESs and the ERP lifecycle. Section 3 will highlight the role of big data and advanced analytics. Section 4 will focus more on the post-implementation stage of the lifecycle and the corresponding problems witnessed within research. Section 5 derives the research problem from the earlier two sections. Section 6 then positions the importance of this research study, followed by Sections 7 and 8, discussing the research methodology and the technical architecture of the IT artifact at hand. These sections would focus on formulating the research as well as the tentative design, triggering the design-science research process used to develop the performance measures leading to scientific results of the modular advanced analytics artificially intelligent agent. It should be noted that one of the main referencing themes within this paper relies on topic relevance to relevant literature. This helps in posing an emphasis towards the persistence of the problem at hand. The paper then addresses the evaluation of the IT artifact in Section 9, and the generalizability dimension of the IT artifact in Section 10. The final two sections, Sections 11 and 12, discuss the concluding findings from the experiment carried out on sample ERP data provided from a large retailer, and the future work, respectively.

## 2. Enterprise Systems

Enterprise systems, also known as enterprise resource planning systems or ERP, are integrated, modular, off-the-shelf systems aiming to control key functional areas within the enterprise such as sales, accounting and finance, material management, inventory control, and human resources [10]. The benefits of using ERP systems are several: seamless information flow, access to real-time data, process-orientation, and improved communication across the enterprise. ERP implementation projects have proved to have high organization and technical complexity, and the human consequences and required changes in business processes are often underestimated [11]. Previous research has identified certain critical success factors (CSFs), which are important for gaining benefits in organizations implementing ERP systems [12]. However, the cost of an unsuccessful ERP implementation can be high, given the number of risks ERP projects experience [13,14]. More importantly, it has also been explained that despite the various research on implementation related CSFs, many of these identified factors are not considered during the post-implementation stage, making ERP projects fail [1].

Research within the field has exhausted the analysis of the ERP system development lifecycle, in which several prominent models appear. Research done by Esteves & Pastor [15] as well as Markus & Tanis [15] mark some of the important lifecycle frameworks. The following snapshot (Figure 1) illustrates the proposed framework introduced where the phases and dimensions are shown.

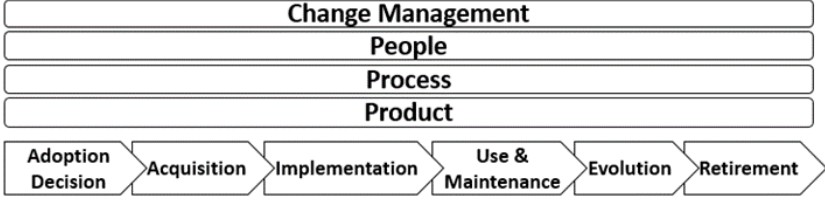

**Figure 1.** The ERP lifecycle [14].

Within this framework, the phases describe the different stages a given ERP system lifecycle goes through in typical organizations (Adoption Decision, Acquisition, Implementation, Use and Maintenance, Evolution and Retirement). On the other hand, the dimensions describe the different viewpoints by which a given phase could be analyzed (Product, Process, People, and Change Management).

Markus & Tanis explains ERPs as a part of ESs [16]. ESs are commercial software packages that enable the integration of transitions-oriented data and business processes throughout an organization. ESs include ERP software and such related packages as advanced planning and scheduling, sales force automation, customer relationship management, and product configuration. The authors claim that the key questions about ES from the perspective of an adopting organization's executive leadership are questions about success. Additionally, it has been claimed that no single measure of ES success is sufficient for all the concerns an organization's executives might have about the ES experience. Consequently, a "balanced scorecard" of success metrics addressing different dimensions at different points in time needs to be adopted, governed by the ES experience cycle phases. Organizations' experience with an ES can be described as moving through several phases, characterized by key players, typical activities, characteristic problems, appropriate performance metrics, and a range of possible outcomes. The phases within this cycle are explained as follows:

*The Chartering Phase* comprises the decision leading up to the funding of an ES.

*The Project Phase* comprises activities intended to get the system up and running in one or more business units.

*The Shakedown Phase* is the organization coming to the adoption state of the implemented ES. Activities include bug-fixing and reworking, system performance tuning, retraining, and staffing up to handle temporary inefficiencies.

*The Onward-and-Upward Phase* continues from normal operation until the system is replaced with an upgrade or a different system. During this phase, the organization is finally able to realize the benefits of its investment. Activities of this phase include continuous business improvement, additional user skill building, and post-implementation benefit assessment; however, these typical activities are often not performed.

Other studies explained some of the challenges an ERP project would encounter during the three phases of the project (pre-implementation, implementation, and post-implementation) [17]. The following Table 1 explains some of these challenges:

**Table 1.** Project challenges over phases [16].

| Pre-Implementation | Implementation | Post-Implementation |
|---|---|---|
| • Defining the right requirements and needs<br>• Selection of system<br>• Selection of vendor<br>• Contract with the vendor | • Ineffective cleaning of data<br>• Inappropriate customization and implementing of module<br>• Project leader spends too much time—conflict with the contract<br>• Problems with allocation of user | • Problems with use of the system<br>• Users avoiding using the system<br>• Problems with module<br>• Problem with support<br>• Problems with user access<br>• Problems with data redundancy<br>• Problems with the customizations carried out during implementation |

Post-implementation problems are the most renowned reasons for general cases of failure being found, even after successful ERP implementation. There is a continuous requirement to rely on external

resources (consultants, vendors, support teams, etc.) for the purposes of either extending the capabilities of the solution through adding additional modules to the ones already implemented, or to follow-up on current existing tickets from users [17]. This has been highlighted long ago in the literature [18]. It has been explained how all preceding factors appearing before the post-implementation phase (Shakedown phase and Onward-and-Upward phase) continue to live, which could be an explanation why is it very hard for ERP projects to survive the post-implementation phase. Although that is the case, many companies do not perform a post-implementation audit of the ERP project to assess and document the achieved benefits [19]. It could therefore be observed that many of the possible earlier problems continue to emerge and contribute to project failures through the years.

These facts lead to the post-ERP depression phenomenon [20]. ERPs usually face very high expectations. However, especially during the immediate time following project conclusion, flaws in the system and newly unveiled bugs tend to dampen expectations, which would consequently dishearten personnel and devastate project managers. That said, post-ERP depression needs to be considered seriously.

In contrast to the difficulties and high failure rate in implementing ERP systems, a cornerstone concept coined in the term CSF has been proposed to develop a more systematic account of ERP implementation, which is useful for guiding implementation management and advancing ERP research [21]. This involves the application of the socio-technical model of system development to ERP implementations. It provides a basis for identifying and classifying CSFs according to model components and interdependencies, developing causal relationships and explaining ERP impact and value added to several organizational dimensions.

Over the past decade, IT investments have grown to become the largest class of capital expenditures in US-based businesses. In the meantime, enterprises consider ESs to be a fundamental tool for organizational distinction, since they integrate organizational systems and enable operational transactions in a cross-functional manner. Successful implementation of ESs has the potential to reduce inventory, optimize production, shipping, labor, and IT maintenance costs, and thus lead to greater effectiveness and a competitive edge [22].

Although ESs have been described as a panacea, there are many reports that run into costly implementations, suffer serious difficulties, and must cope with severe maintenance problems along the implementation lifecycle [22].

Previous research is documenting organizational issues and problems connected with time over-runs and lack of alignment of organizational structure with the adopted ES. Meanwhile, practitioners from industrialized countries recognize the implementation and post-implementation complications linked to ESs as lack of user engagement. ES implementation projects suffer from interdepartmental disagreements, insufficient training, and system customization problems [23].

## 3. The Role of Big Data and Advanced Analytics

Big data has become important as many organizations have been collecting massive amounts of data which can contain useful information about given problems i.e., ERP log analysis, through the ability of employing highly scalable architectures, surfaced through typical Hadoop architectures. This could in turn enable detailed business analysis and decisions impacting existing investments and future technology.

Big data as a technology unlocks the powers of unstructured data. This paves an open highway for text analytics in big data to unlock the transformations sought by the technology. From unstructured data, the process of extracting information and facts is called text mining [24] and knowledge discovery in text (KDT). This is a rising field of information recovery, statistics, machine learning, and computational linguistics based on the fact that most big data is actually unstructured text [25]. There are many approaches of text mining. The major approaches are the keyword-based approach and the information extraction approach. Text analytics solutions use a combination of statistical and content analysis to extract information. Statistical analysis is carried out on text at various dimensions such as

term frequency, document frequency, term proximity, and document length. Content analysis on text takes place at different levels:

*Lexical and syntactic processing:* recognizing tokens, normalizing words, language constructs, parts of speech, and paragraphs.

*Semantic processing:* extracting meanings, name entity extraction, categorization, summarization, query expansion, and text mining.

*Extra-semantic features:* identify feelings or sentiments (feelings, emotions, and mood).

*Goal*: Dimension reduction [26].

Correspondingly, state-of-the-art technological paradigms catalyze analytics models and algorithms for information mining, machine learning, and ultimately artificial intelligence (AI). AI dates to 1950s, explained in Turing's intelligence test [27]. However, it was not until 2006 that the fruit of AI was harvested, when Geoffrey Hinton, Osindero, & Teh [9] proposed the deep architecture of an unsupervised greedy layer-wise learning manner, creating the concept of deep neural networks. Deep-learning algorithms extract high-level complex abstractions as data representations through a hierarchal learning process [28]. Complex abstractions are learnt at a given level based on relatively simpler abstractions formulated in the preceding level in the hierarchy. These abstractions enhance the process of feature engineering and extraction, whereby the ability to construct features and data representations from raw data is elevated. Many practitioners think of deep-learning architectures as the natural evolution of ANNs in which intermediary hidden layers grow out of proportion to the original ANN models. Several commanding breakthroughs have been introduced through applications of deep convolutional networks (one of the types of algorithms introduced under the deep-learning paradigm) since the early 2000s, with great success in detection, segmentation, and recognition of objects and regions in images. These were all tasks in which labeled data was relatively abundant, such as traffic sign recognition, segmentation of biological images, and detection of faces.

## 4. Research Gap: ERP Post-Implementation Problems

ERP systems have been criticized for not maintaining the return on investments (RoI) promised. Sykes et al. (2014) claims that 80% of ERP implementations fails. Also, 90% of large companies implementing ERP systems failed in their first trial [3]. It has also been reported that 50%–75% of US firms experience some degree of failure. Additionally, 65% of executives believe ERP implementation has at least a moderate chance of hurting business [29]. Three quarters of ERP projects are considered failures and many ERP projects end catastrophically [30]. Failure rates are estimated to be as high as 50% of all ERP implementations [31]. In addition, as much as 70% of ERP implementations fail to deliver the anticipated benefits [32]. Still, many ERP systems face resistance and ultimately failure [1]. That been said, there are two main critical phases that might lead to the failure of an ERP project. The implementation as well as the post-implementation phases mark the two main phases in a given ERP lifecycle where many of the organizations might experience failure. The two phases include similar activities and involve similar stakeholders.

Post-implementation lack of success within small- and medium-sized enterprises (SMEs) as well as large enterprises reflects how the human factors of failure dominates the scene of malfunction. This would then lead to what is known as a *lack of continuous*, *productivity decline*, or *early retirement*, which causes performance dips within given organizations. The main themes that had furnished a foundation for ERP post-implementation failure within SMEs are: (1) workarounds; (2) lack of training; (3) incentives among resellers and implementation consultants; and (4) lack of culture openness within the company, where every one of those pillars would lead to another [33]. This foundation would surely lead to the inflation of the hidden costs of a given ERP project, which diminishes the expected RoI [34].

ERP systems usually impose business process standardizations through routines, roles, and data with the aim of optimizing the business processes within the organization. Unfortunately, many users, when faced with problems that break the flow of daily business, would adopt a workaround [35].

A workaround employs the functionality of the ERP system by deviating from the intended design, achieving the required functionality. Although workarounds do not change the ERP system or the business processes, workarounds usually have dire consequences on productivity and often degrade process excellence and increase the possibility of making errors [36]. The typical aim would be to overcome the performance dip through finding workarounds to get the company back to normal operations as fast as possible. The concept of a workaround is defined as "computing ways for which the system was not designed or avoiding its use and relying on alternative means for accomplishing work" [37]. Workarounds, as earlier pointed out by Gasser [37], and researched by Alter [38], usually lead to errors that eventually cause many other problems to the underlying database that interferes with the work processes of other employees. The work required to rectify these database errors are considerably time consuming, which henceforth would contribute to the elongation of the performance dip within the organization. This is observed by analyzing the tendency of "drifting", which describes the discrepancy from intended use towards context-based user patterns [39].

Additional reasons that contribute to ERP post-implementation failure and enforce the adoption of workarounds would be lack of training [40]. In many cases, once an ERP is implemented, the vendor works with employees to educate them on how to use the system and follow best practices. Unfortunately, many organizations fall into the trap of inadequate and low-quality training. Through an inability to efficiently coordinate the flow of the training course, lots of discussion with curious and inexperienced users is triggered. That would then leave the trainees with hectic and uneven learning.

Through inabilities to build up internal proficient competencies within an organization, external consultants are brought to the table to transform failure into success. To continue the chain of consequences leading to post-implementation failure, there is a requirement to hire a subject expert to both rectify and discontinue adoption of workarounds. Resellers or vendor partners would typically be at the top of a list to handle these tasks, through deploying their experienced implementation consultants. In such situations, consultants could make it difficult for the hiring organization to dispense their expertise to secure the revenue flow. One method would incur a deficiency on the documentation of the system modifications being made to rectify the ramifications of the workarounds. Even though consultants are very important to ERP implementations, consultants would often follow their own agendas, which might do more harm than good to the organization [41,42].

It has been identified through research that many different aspects cause many ERP projects to fail, on top of which lack of organization learning comes into perspective [43]. The research community recognizes the essential role of organizational learning and the relationship between an ERP system across the ERP implementation lifecycle. It is worth mentioning that the theory of organizational information processing (TOIP) suggests that an organization's fundamental task is to process information concerning uncertainty in the environment and to make decisions to deal with it [44]. If the uncertainty could not be settled effectively, it can translate into volatility of firm performance, causing firm risk. Firm risk degrades ERP business value as a direct trigger by post-implementation bottlenecks.

Furthermore, in the context of a large enterprises, several factors affecting the post-implementation phase complements the earlier discussed factors affecting SMEs [45]. Customized workarounds and post-implementation training were two common factors between SMEs and large enterprises. It had been noted that the higher a customization or a workaround is adopted, the more the negative impact is on the ERP system. Additionally, training is discussed to be a crucial factor keeping the ERP system successful. Moreover, among the key factors hindering the success of an ERP post-implementation are:

- *Lack of Executive Management Support:* Top management support is a critical recurrent factor which affects every stage of the ERP lifecycle. Enormous support given by top management pays off in the post-implementation phase, where users get the opportunity to use a system that satisfies their expectations.
- *Initial Post-Implementation Benchmarking:* As stated by an independent consultant, "initial post-implementation benchmarking is the measurement of user expectations with actual

performance, at the initial stage of the post-implementation stage". This practice includes setting milestones and benchmarking parameters and estimating expected versus actual.

- *Change Management* Illustrate the importance of proper change management initiatives to achieve the true benefits from an ERP system.
- *Maintenance of ERP* According to past literature, ERP maintenance is the process of updating the ERP to meet changing user requirements. Almost all the case examples have made the effort to maintain the ERP. Help-desk support and ERP support team teams were used as criteria to determine the existence of ERP maintenance. Adherence to any of the above criteria determines that a particular case has carried out successful ERP maintenance.
- *Introduction of Additional Features at the Post-Implementation Phase:* According to an independent expert, this is a practice where vendors initially provide the basic ERP system with the basic set of functionalities. The additional features mentioned in this practice are about the value-added features such as enhancements to the user interface and attractive templates.
- *Success of Pre-Implementation* Pre-implementation success directly impacts on the post-implementation success.

The above pointers open discussions to the very important socio-technical impact on any ERP project. Lack of transparency caused by workarounds among the socio-technical stakeholders is one of the main reasons many projects reach the point of failure. That is why application stewardship or application governance had been discussed in an attempt to increase the value of applications in the hands of users [46]. This theoretical socio-technical framework explains the relationship models between IT professionals and user organizations, with a goal towards increasing the functional outcomes of a given application.

Despite the huge value posed by application stewardship in mapping these activities to the bigger IT governance frameworks to preserve application values, the cost of establishing a governance board is typically high, especially for SMEs [47]. That would then expose the gap for how augmented ESs are able to provide a rather cost-friendly approach.

## 5. Research Problem

In this research, we address the problem of ES post-implementation failures [1,48,49]. We seek to answer the research question of how advanced analytics could help ESs during post-implementation activities. The idea is to design and develop an advanced analytics engine (AAE) that seeks to be artificially intelligent. The role of AAE is to understand the implementation environment, detect issues, predict system failures, and suggest and verify solutions. In this way, the AAE will enable ESs to rely less on vendors and third-party Service Level Agreements (SLAs). This will enable organizations adopting ESs to achieve their implementation targets. The AAE will employ advanced analytics techniques, machine learning, and data mining. These domains have been proven to have a strong presence through academic as well as industrial arenas, via robust classification and prediction methods. This will help make ESs self-moderated, by reducing the reliance on vendor support. The idea is to bring analytics to transactional systems, with the objectives of using analytical capabilities to enhance the functionality of these systems.

The AAE will act towards helping ESs in two phases:

Phase I: in this phase, the AAE will analyze the environment, which mainly includes business processes via their business blueprint, and examine where data insights can be useful to support those processes. Also, support tickets, product documentation, and related email communication needs to be analyzed and understood.

Phase II: in this phase, the AAE will provide analytical capabilities related to failure prediction, solution confirmation, and full integration. Roles which were supposed to be undertaken by humans—either support staff or SLAs—are supposed to be automated. Added to that, the availability of the ES will be further enhanced.

## 6. Importance of the Study

With the abundance of data-creating technologies, businesses are gathering data generated at high velocity from various sources such as the Internet of Things (IoT), sensory and social media, and in many different formats such as structured, semi-structured, and unstructured. Such abundance is coupled with ubiquity and value, whereby humans and businesses alike are leaving a digital trace [50]. Today's data are limitless in terms of size, where each record is represented by a huge number of attributes that have less clear dependence compared with other records [51]. Therefore, a combination of data from many diverse sources represents a new challenge, and at the same time an opportunity for business [52]. Accurate data comes at a cost, demanding businesses to study the trade-off between the benefit of more data and the cost of acquiring it [53]. In the meantime, advancements in analytics is making computing cheaper, and AI tools, such as machine learning, are lowering the cost of prediction. These trends have put further pressure on organizations to sense and respond faster and be more competitive. AI techniques now extract more value from data. Therefore, this study comes in a timely manner to make use of data abundance, taking it as an opportunity, using machine-learning techniques and AI as methods to enhance ES post-implementation performance.

Despite their pervasiveness, ERP systems suffer from a serious concern regarding the failure of ERP implementation [54]. Sykes et al. [55] claims that 80% of ERP implementations fails. It has also been reported that one of the extremely important factors in ERP post-implementation is ongoing vendor support. Therefore, this study sets out to use machine learning and AI to build an AAE, which is making use of the sheer amounts of data available to businesses to make ESs more resilient to failure and hence reduce total reliance on vendor support. The use of the AAE is going to be a game-changer, as it changes the way businesses and those implementing and running ERP systems rely on external support from vendors in financial and technical terms. Staehr et al. [56] examined post-implementation periods in four manufacturing companies and proposed a framework for understanding the benefits that businesses achieve from ERP implementation. The framework consists of themes where support is one which could leverage the ERP system. According to Oseni et al. [57], maintenance and upgrade costs are ever-present in the yearly ERP budget. Client companies of some of the larger ERP vendors, such as SAP and Oracle, can be charged hundreds of thousands if not millions of dollars in maintenance, annually. The ongoing costs of maintaining, modifying, and updating an ERP system is substantial, even for mid-sized organizations. Thus comes the importance of such a study. Organizations adopting the AAE will benefit via dropping their reliance on external parities, lessening operation costs and reducing system failure times. Altogether, it will increase the availability of ESs and accordingly optimize business operations.

## 7. Research Methodology

The AAE would work on defending ERP implementations from post-live problems. The AAE, to be able to generalize its usage, will be developed via a detailed framework that works on defining the integration of the analytical points between the ERP and analytical algorithms. For this purpose, design-science research (DSR) is to be adopted [44].

Our process aims to follow the guidelines explained by Hevner et al. [58] to build the AAE.

On the other hand, the DSR process model explained is to be adopted through the six research steps to be used to devise a complete artifact, leading to the solution [59]. The process, Figure 2, begins by identifying the motivational problems within the topic, which is something that has partly been explained in this proposal. Second, it identifies the suggested objectives of the solution, which explains how our AAE would work to help ERP implementations during post-implementation phases. During the third step of the research, we would be looking into the design and development of the artifact to serve the purpose defined in the objective. Fourth, the artifact is to be evaluated and tested by examining how the developed artifact performs against an actual ERP case study. Finally, the conclusion step of the research journey comes through publications as well as industrial events. It should be noted that this is an iterative process model where several outputs are produced at the

end of each iteration. In this case, the first iteration cycle produces the proposal for the research project, while the second is in operation until (n-1) cycles are used to extract the tentative design, the artifact design, and performance measures. This would then end with the results observed during an experiment to test the IT artifact.

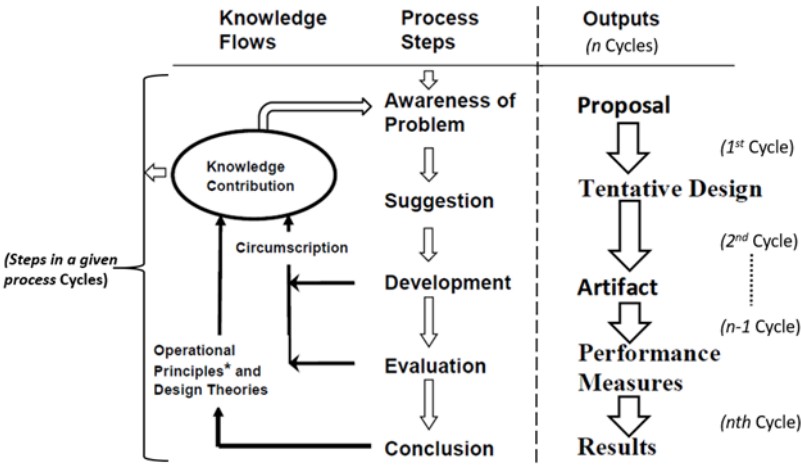

**Figure 2.** Adopted DSR process model: amended illustration of the cyclic research process from Vaishnavi and Kuechler [59] and how outputs mature from these cycles.

*Artifact Design and Development*

From the earlier sections, it could be deduced that the artifact in discussion is the AAE, which works on overcoming ERP post-implementation challenges. By design the AAE should be application-agnostic, ready to modularly integrate with any ERP for analyzing the dimensions of challenges. The tentative design technology components should include data feeds to be used for training the analytical models used to train the AAE. Analysis of the challenging dimensions should take place through employing advanced analytical techniques carried over from data feeds extracted from support tickets, emails, application back-end repository, and application logs. Text mining should be used to build contextually related resolutions to given post-implementation challenges queried by end users. The training dataset should enable the agent to learn from previously resolved support tickets through analyzing historical data extracted from the different source collections. Deep-learning architectures could then be used to analyze the sources of data, furnishing a foundation working on avoiding post-implementation hazards that lead to organizational process failures.

As illustrated in the diagram shown in Figure 2, the design and development phase should address a continuous feedback process that works on the development of the artifact at hand and the realization of the sought results. Coupling this process with the aforementioned tentative design components should obtain conclusive results leading to the artifact. The design guidelines would then be explained as follows:

- *Data Selection:* The first step of the process is defined in the action of determining appropriate data types and sources to be used for collecting data either in batches or in real time. This step therefore defines the types of instruments to be used for collecting the data without compromising the integrity aspect.
- *Data Pre-Processing:* Secondly, understandability of the data is one characteristic that needs to be maintained to enable accurate and efficient execution of the various data-mining techniques. Often, big data is incomplete, inconsistent, and/or lacking certain behaviors or trends, and is likely to contain many errors. Data pre-processing prepares data for further processing through employing data-transformation techniques, data cleaning, data integration, data reduction, and data discretization.

- *Data Transformation:* To gear up the data with a trait of usability, data-transformation methods would be employed to read/parse the data in its original and then translate the data to be used by the data-mining techniques.
- *Data Mining:* To produce particular enumeration of patterns or models over the data, this step takes place as part of the advanced analytics discovery process by applying data analysis and discovery algorithms.
- *Interpretation and Evaluation:* Interpreting the analyzed and discovered patterns works by producing understandable relevant information that should be used to explain the data population or corpus. This step makes up the feedback loop to previous steps within the discovery process, as well as automated event trigger as a result of the entire process. Visualization of the identified patterns furnishes a base for manual user intervention.

## 8. The Artifact: Technical Architecture

For some, an artifact could be a system or a system component, while for others, artifacts need to be in a form of theory or theory components [60]. According to Baskerville et al. [61], "if an artifact is novel and useful, then it necessarily contributes to design knowledge". Additionally, "two common types of contributions are defined as research outcomes from a DSR project—design artifacts and design theories" (p.359).

Adhering to the cyclic research process from Vaishnavi & Kuechler [59]; our artifact explains the awareness of the process steps as follows: the artifact at hand is one that works on discussing the encapsulated method used to solve chronic post-implementation problems facing businesses running on ESs, hence realizing the process of problem awareness.

The suggestion process explains how the artifact is to be composed out of three main components, which transfers the problem into the confirmed solutions. Correspondingly, these components are:

- *Problem Identification*: this component identifies the problem once it is occurring. The identification could be manually handled via user input, or automatically through an agent.
- *Problem Resolution***:** this component captures related relevant data to the problem and applies advanced analytics and machine-learning techniques to find a solution to the problem. This component represents the core idea of the AAE. The AAE Problem Resolution is a composite component which is uniquely characterized by: (1) use of big data such as support tickets; forums; documentation; and business blueprint; (2) use of advanced analytics such as text analytics, business process intelligence, and deep learning. Although techniques are not new, the algorithmic approach of using such techniques towards solving the problem is unprecedented; and (3) use of state-of-the-art techniques such as in-memory computing.
- *Solution Confirmation***:** this component ensures the solution can resolve the problem. This confirmation could be manually sourced via user, or automatically via simulation agent.

During the first iteration output of the DSR model, the following tentative high-level design explains the initial development state of the artifact at hand, which would then be discussed through the following sections. Correspondingly, Figure 3 explains how the designed AAE should sit on top of any given ES to enhance its functionality. Automatic problem identification and solution confirmation is addressed by the AAE both to spot problems and to propose corresponding solutions. Additionally, the illustration shows how the purposefully designed spoken dialogue system (SDS) or chatbot helps users to manually interact with the AAE for problem identification and solution-confirmation purposes. System failure would then be safeguarded through the ability to analyze text data feeds coming from support tickets, emails, log files, business blueprint (BBP), and forums. That said, this high-level design should set the guidelines for the development of the technical architecture. These guidelines are to be later followed by evaluating the efficiency and accuracy of the architectural framework through a set of performance measures.

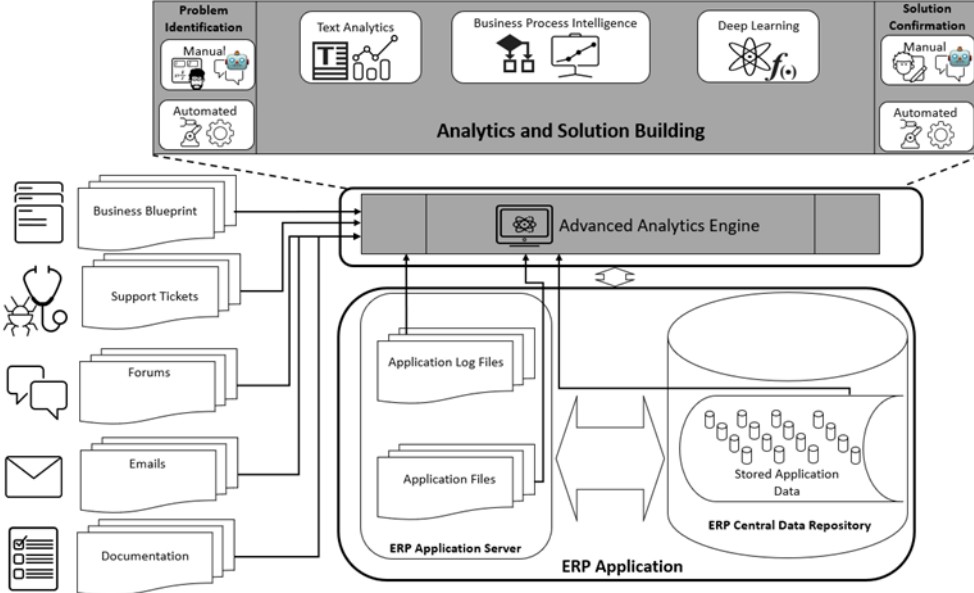

**Figure 3.** Illustration of the system design including an advanced analytics engine (AAE).

### 8.1. System Modularity

It is a known fact that many ERP vendors are shipping different types of systems for specific purposes serving a digitally governed business process for efficiency purposes. Correspondingly, modular design of the AAE is one that enables it to interact with any ERP. This modular design is one that lies in the ability of the system to integrate with the data sources regardless of type, format, or technology. Information integration with the sources is governed by the analytical data model used by the agent to carry out text analytics, deep learning, and business process intelligence mapping against the system at hand. Automatic feedback takes place between the AAE and the ERP for automated problem identification and solution confirmation. Depending on analytics accuracy, the AAE would be able to cater for requirements posed by the chatbot to be used by the users for manual problem identification and solution-confirmation purposes. Given the level of modularity of this AAE, both cloud and on-premise deployment models could be achieved, depending on organization preference.

### 8.2. System Interoperability

It can be observed based on the system design that a resource-intensive architecture is required. A big-data architecture is then employed to host the AAE to support the parallel multi-threaded analytical executions over the sheer amounts of text data coming from the many sources. This, by turn, defines the interoperability model of the system, whereby a resource-intensive architecture residing on a Hadoop cluster is formed for text pre-processing, and for topic modeling through multi-document summarization methods.

### 8.3. Data Sources

Through the ability to process various large and fast data inputs, as part of the AAE architecture, a big-data manifest materializes. Textual data sources spanning a considerable number of touch points should be processed by making use of several text analytics techniques. However, it is worth mentioning that these text sources are ones that essentially enriches the understanding of a given obstruction point within the post-implementation process, hence establishing the basis for a rather accurate automated problem identification. Through the ability to combine structured information specific to the ERP application at hand, this setup could be achieved. That said, one of the main focal points would be to sensibly indicate quintessential sources of data dissemination, which would correspondingly maintain the modularity and interoperability traits of the architecture as well as

deliver a holistic digital understanding of probable problematic areas leading to an automated solution confirmation. Henceforth, the unstructured text-based sources that are part of the architecture are explained as follows:

- *Business Blueprint:* possibly the most vital document explaining and setting the implementation guidelines for a given implementation process. This document sets a common strategy of how the organizational business processes are to be selected and implemented. Through explaining the details of the organizational units, master data, business scenarios, business processes, and process steps, the defining key aspects of the ERP implementation process could be realized.

- *Documentation:* Typically, any business solution comes with product documentation manuscripts that explain the code of operation of the system at hand. These documents usually touch base on the different components being used to make up and formulate the solution. These documents are usually the main source of information for technical insights defining the application. Hence, these documents are ones that should construct the digital understanding of what is the purpose of the system, how to manage the code of operation, and how to possibly rectify typical problems.

- *Support Tickets:* A support ticket is traditionally a communication trail-signaling problem with the solution between the technical support engineers and functional/technical consultants working on the system. These textual artifacts describe given problems and corresponding probable solutions to the problem being raised.

- *Forums:* With many of the vendors, technical forums are being created to furnish a social and technical platform where people with technical expertise interact with one another to flag bugs, problems, etc., and the corresponding methods of resolution to the problems. These forums usually possess the trait of technical richness in defining and resolving the problem through collaborative detailed explanations of the resolution tracks.

- *Emails:* Within any given organization, email communication makes up one of the most important means of communication that could be used to define a given problem with a given system. These textual sources of information, if analyzed, could narrow down the margin of error in identifying a given problem with a given application.

That said, through the ability to contextually move between these different textual artifacts suggest a rather holistic understanding of a given problem could be reached, leading to corresponding automated resolutions. On the other hand, structured data points that could be directly extracted from the ERP could be also used to verify a given problem by associating the identified problems of the earlier unstructured textual sources with the data hosted within the ERP central back-end repository as well as the application logs. Once these contextual associations take place between the different data touch points, a corresponding solution could be also confined for the final process of resolving the problem, leading to the final stage of solution confirmation.

### 8.4. Technical Architecture

To be able to carry out the different processes of problem identification, analytics/solution building, and solution confirmation, advanced analytics and data processing (batch and real-time) technical capabilities are required for the sought-after prototype. The following diagram (Figure 4) explains the tentative design of the technical architecture dissecting the anatomy of the technological products being used to furnish the basis of execution.

This technical architecture relies on several Hadoop architectural interfaces within the big-data infrastructure (Hive/Impala, Spark, etc.) to facilitate the process of data dissemination between the different data points to be used in formulating the AAE solution. Data selection, pre-processing, and transformation is taken care of through making use of the different interfaces appearing within the architecture. After all, the entire Hadoop architecture is one that has proved to be strong towards harnessing unstructured and semi-structured data pertaining to these three steps within the discovery

process. To explain the compositional build of this, the above figure describes the technological attributes making up this technical architecture.

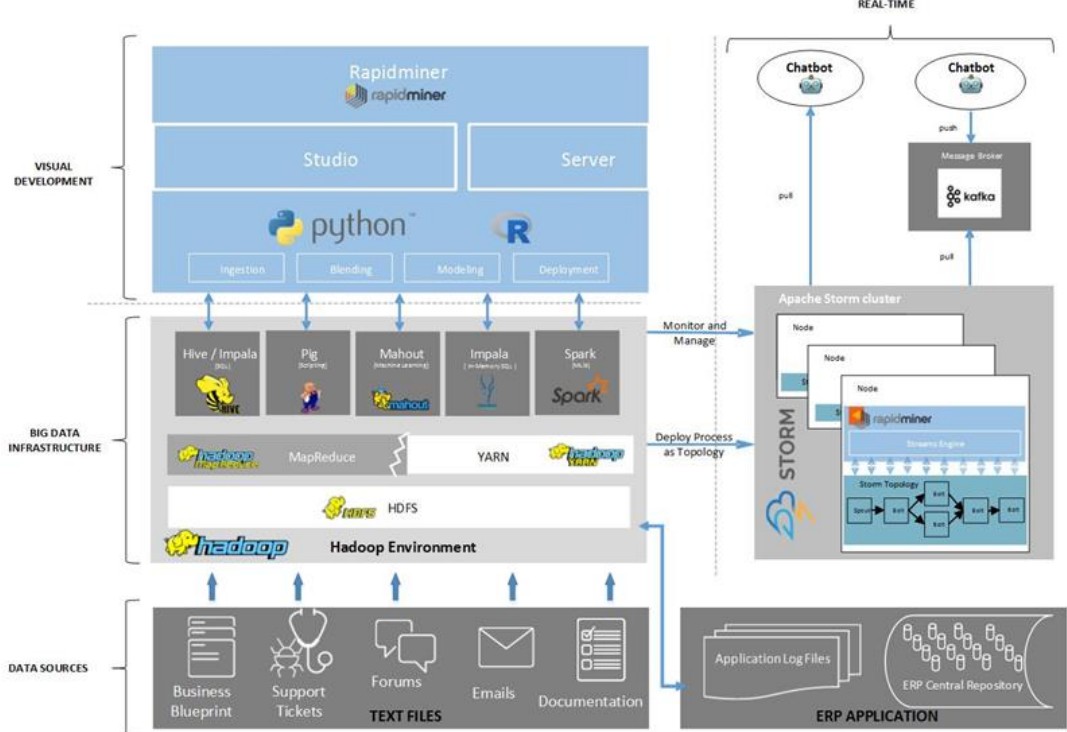

**Figure 4.** Illustration of the technical architecture design of the artifact showcasing batch and real-time data dissemination and processing.

One of the main components within this prototype architecture is the visual development platform that is being used to govern and facilitate the processes of execution on either the data ingestion or advanced analytics processes. Therefore, RapidMiner acts as the analytics umbrella platform used to facilitate the different steps of the discovery process. Through the openness trait of this platform, additional open-source languages (Python and R) are to be used to accelerate the statistical capabilities of the architecture, as well as sophisticated advanced analytics methods expressed in the form of deep learning, for instance.

Keras, among others, is one of the prominent high-level neural network libraries, written in Python and working as a wrapper to other deep-learning tools. Similarly, R has many libraries that excel at statistical analysis and prediction.

The abstract mapping between the conceptual architecture and the technical architecture is illustrated in Figure 5. Apart from the common data sources layer on both diagrams, it could be observed how analytics and solution building, comprised of three pillar processes making up the composition of the AAE, is mapped to the visual development layer of the technical architecture. This mapping indicates how RapidMiner, as the data-mining development tool, is used to furnish a development base for the three processes of the AAE. Through combining the Out-of-The-Box (OOTB) capabilities of the tool with open-source languages such as Python and R, the entire discovery process could be addressed.

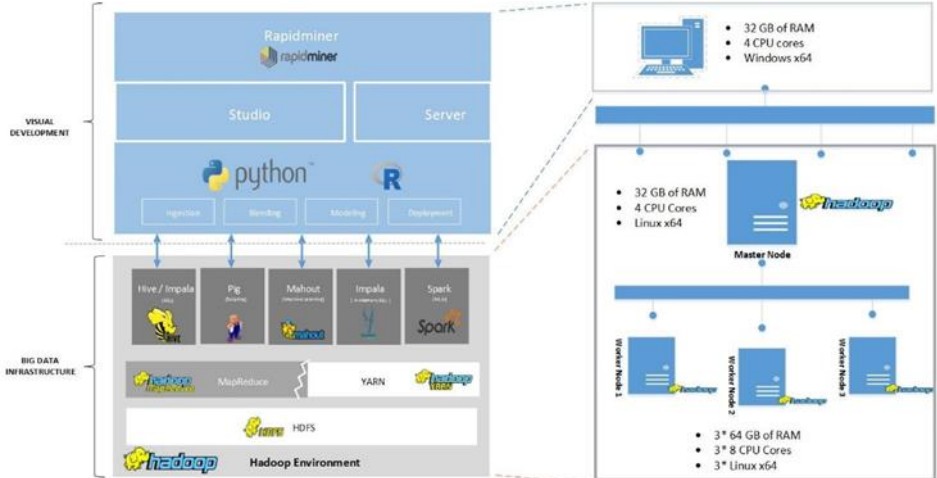

**Figure 5.** Illustration of the technical architecture design of the mapping to hardware resources.

*8.5. Analytical Techniques*

The AAE artifact works through three pillar components—problem identification, problem resolution through advanced analytics and solution building, and solution confirmation. The ultimate purpose of these components would be to ensure that the solution can resolve a given ERP post-implementation problem leading to safeguarding ERP implementations. Given the abundance of textual data being ingested into the framework, extractive and abstractive text summarization techniques would then define the core of the problem-identification component, helping to condense these large amounts of textual data for a case identification. Text summarization techniques help achieve the following key aspects: (1) removing several redundancies with corpuses, which leads to efficiently executing the analytics algorithms without wasting time on processing repetitive data; and (2) removing data that is not essential for the problem-identification process within given documents/corpuses [62]. Therefore, the text summarization techniques firstly pre-process the relevant sentences while maintaining the coherence of the document/corpus. By combining extractive and abstractive text summarization techniques, term frequency-inverse document frequency (TF-IDF) problems could be identified automatically. Within the context of this problem-identification process, rare occasions of manual human intervention are expected to occur. This expectancy of the manual intervention would be defined by the accuracy measures pertaining to the algorithms used within this process.

Once the given problems are identified, the process of problem resolution begins. Advanced analytics techniques define the core of the solution-building component. Motivated by the requirement to analytically close on resolutions to the identified problems, information extraction (IE) algorithms are used [24]. Entity recognition (ER) is also employed where entities are found within the text and classified into predefined categorical solutions. Supervised ER techniques such as decision trees, hidden Markov models, conditional random fields, etc. are applied in a combination with unsupervised learning techniques such as clustering algorithms. On the other hand, relation extraction (RE) algorithms work on finding and extracting semantic relationships with pre-identified problems and corresponding resolutions from the pre-processed ingested text from ERP blueprints, documentation, support tickets, emails, and forums. Deep-learning algorithms also play a pivotal role in the process of IE. Given the powerful capabilities of these algorithms, deep learning helps in lifting the accuracy of the IE process.

*8.6. Building the AAE Architecture*

Throughout the following sections, a set of pointers will be discussed to explain how the tentatively designed technical architecture is being transformed into a physical infrastructure. This physical infrastructure should be capitalized on for realizing the physical prototype (the DSR artifact). These

sections follow the DSR process model illustrated in Figure 2 [59], for which at this stage the process outputs evolves into explaining the artifact product based on the earlier explained tentative design. It should be noted that to realize a fully operational physical prototype, additional DSR iterations are required to explain the logical data structures to be used in influencing the physical prototype from an analytical perspective addressing the three components of the AAE (problem identification, problem resolution, and solution confirmation).

The tentative DSR artifact design has paved the way towards building and running the realized big-data infrastructure. The infrastructure build has been constructed to retain the intactness of the state-of-the-art technologies to be consumed for the purpose of assimilating large, various in type, and fast unstructured and semi-structured data. Correspondingly, changes to the methods and anatomical composition of building the infrastructure has been embraced, for which an illustration to the changes are highlighted in Figure 6.

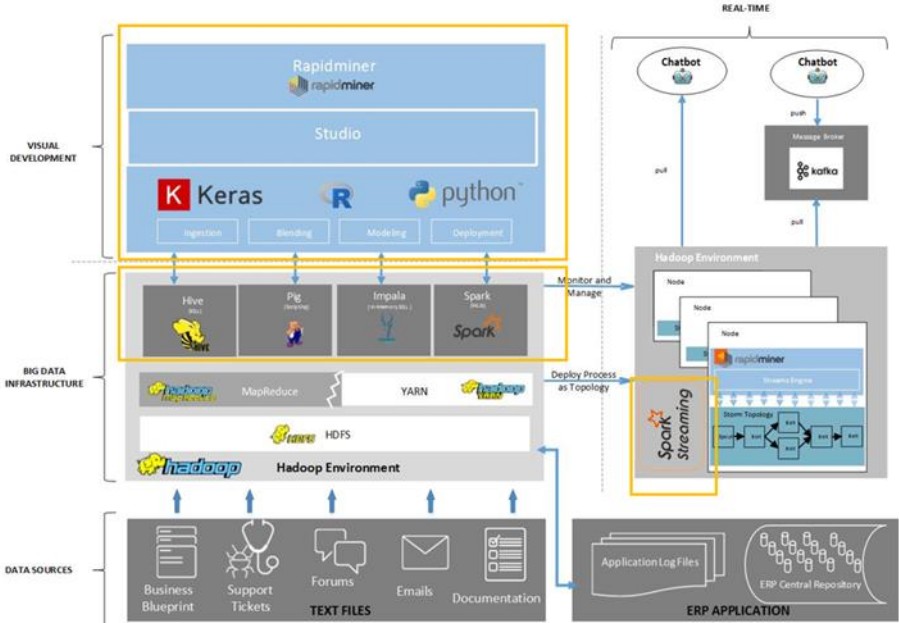

**Figure 6.** Highlighted in orange, anatomical changes to the AAE artifact.

Firstly, the current infrastructure has been cloud hosted. The reserved hardware infrastructure hosts 4 cloud compute machines and one additional virtual machine hosted on the public cloud and is accessible remotely.

It should be noted that the upfront relative increase in the computing resources has been founded only on the ability to leverage the public cloud provider template resources, and not based on performance benchmarking rigor.

On the other hand, among the changes comes the requirement to avoid the impact of virtualization technology overheads [63,64], on the already virtualized machines hosting the infrastructure. Likewise comes the elevated levels of accompanied administrative maintenance of the virtualization infrastructure on the guest virtual machines versus the hosted cloud machines. That said, the approach of directly installing the big-data technologies on the cloud compute servers has been adopted. In this case, Cloudera Hadoop distribution is the distribution being installed. This Cloudera installation has also resulted in a corresponding change in the real-time element of the architecture given an out-of-the-box streaming component that gets installed automatically, for which this component is Spark Streaming, substituting Apache Storm.

The advanced analytics layer is manifested under the visual development platform. This is built with an aim of matching the underlying powers of the massive parallel processing engines hosted on the Hadoop architecture. RapidMiner is the general advanced analytics/data-science visual development

platform that is used for the purpose of interfacing with underlying big-data infrastructure [65]. It is an open-source graphical user interface that facilitates running Hadoop jobs. One of the key features of the tool would be the API extensions that enable the integration of Hadoop with other user interfaces and state-of-the-art data-science platforms. This feature has influenced an anatomical change in the composition of the advanced analytics/data-science components taking part in the structural build of the platform powering the processes of AAE problem identification, problem resolution, and solution confirmation. The change discusses incorporating state-of-the-art languages and API in the form of Python, R, and Keras, as a substitute to Mahout.

The decision to include Python, R, and Keras as the main components as opposed to Mahout is one that is influenced primarily by the vast capabilities of these two languages, with all the pre-packaged libraries made available for machine learning. The eminence of these languages is also evident from both an academic and industrial view. Recent KDnuggets software poll carried out by Shapiro [66] indicates that R and Python are the two most popular data-science tools partly making up the composition of our platform.

The complementary addition of Keras was influenced by the fact that RapidMiner, which is the key platform in our data-science development layer, has out-of-the-box extensions with Keras. While Keras is generally known to be an easy-to-use Python-based deep-learning library, the augmentation with RapidMiner adds an elevated level of simplicity and puts an emphasis on the system interoperability and modularity traits of our artifact.

## 9. AAE Evaluation

As per Venable, et. al. [67], evaluation is a pivotal activity in conducting DSR. In line with their seminal paper, we classify the AAE as a tool (product), rather than a process, and as technical instead of purely social. We have also taken a naturalistic approach to evaluation by relying on real data and real users. Meanwhile, since the AAE includes mathematical and statistical machine-learning models, so it leans itself more towards the artificial evaluation. Therefore, we have set up a lab experiment on which the AAE was built and tested, as a type of ex ante evaluation, in addition to a case study with a retailer, as a type of ex post evaluation. See below for more details.

In this section, we explain how the AAE is going to support problem identification, solution, and verification of the solution via application. We have contacted a case study company working in retail. The reason for the convenience sample is attributable to the fact that the retailer has been working on an ES for more than 10 years; they have run through three full implementations, and they have substantially large implementation, including over 400 users and 80,000 articles.

### 9.1. Data Acquisition

We have obtained four datasets: Support Tickets, Business Blueprint, Application Failures, and application log files.

*First—Support Tickets:* in our attempt to implement the AAE to support post-live activities, we have acquired the support tickets of ten years 2008–2017. For the first eight years the retailer—support ticket provider—was running on Oracle ERP, whereas for the last two years the retailer was running on SAP ERP. Each ticket includes the following attribute structure: {ID, date, problem description, Requester, Solution, and Solution Authority}. See example below example:

| | | | | | |
|---|---|---|---|---|---|
| 8000037002 | 01102017 | Both the Inbound and Outbound are not working | Finance Department | The Java server is down and the service needs restart | Application Department |

*Second—Business Blueprint:* added to that, we have also acquired the business blueprint documentation of all implemented modules at the retailer. That includes: sales, financials, purchasing, and human resources. See example below, which shows accounts receivable—AR—process steps:

| PROCESS STEP DESCRIPTION | | | | |
|---|---|---|---|---|
| ID | Process Step Description | Ref. to Requirements list If Exist | Business role | Transaction code |
| AR-01 | Create Customer | | AR Accountant | FD01 |
| AR-02 | Change Customer | | AR Accountant | FD02 |
| AR-03 | Display Customer | | AR Accountant | FD03 |
| AR-04 | Display Changes for Customer | | AR Accountant | FD04 |
| AR-05 | Block/Unblock Customer | | AR Section Head | FD05 |
| AR-06 | Set Deletion Indicator for Customer | | AR Section Head | FD06 |
| AR-07 | Credit Memo – Enjoy Transaction | | AR Section Head | FB75 |
| AR-08 | Credit Memo - General | | AR Section Head | F-27 |
| AR-09 | Park/Edit Credit Memo - Enjoy Transaction | | AR Accountant | FV75 |
| AR-10 | Credit Memo Parking - General | | AR Accountant | F-67 |
| AR-11 | Parked Documents-Post/Delete | | AR Section Head | FBV0 |
| AR-12 | Parked Documents-Change | | AR Accountant | FBV2 |
| AR-13 | Parked Documents-Display | | AR Accountant | FBV3 |

*Third—Failures:* the failures dataset was acquired showing three structures: user structure, failure meta-data, and failure detail. For the failure record, see the below example, which includes {date, failure authority, failure scope, type of failure, planned/unplanned, duration, action taken}:

| 21052017 | Application Department | SAP ERP | Application Failure | Unplanned | 1.25 min | Release core files |
|---|---|---|---|---|---|---|

*Fourth—Application Log*: the application log of the ERP is an important source of understanding failures. Therefore, we have obtained application log to better understand failures. ERP system failures are not so frequent, but are often very costly. Such failures can have an unfavorable impact on the business of any organization. The ability to accurately predict a forthcoming failure adds an essential functionality to ESs. See example below, which shows the subset of the parameters taken on regular basis:

| SAPLSMTR_NAVIGATION | Transaction PC00_MEG_CALC Started | PC00_MEG_CALC |
|---|---|---|
| RPMENUSTART00 | Report RPMENUSTART00 Started | RPMENUSTART00 |
| HEGCALC0 | Report HEGCALC0 Started | HEGCALC0 |
| SAPLSMTR_NAVIGATION | Transaction ZWDBM Started | ZWDBM |
| ZPOS_WDBM | Report ZPOS_WDBM Started | ZPOS_WDBM |
| SAPMHTTP | Logon successful (type=H, method=P ) | H |
| SAPMHTTP | Logon successful (type=H, method=P ) | H |
| SAPMHTTP | Logon successful (type=H, method=P ) | H |
| SAPMSYST | Logon successful (type=A, method=P ) | A |
| RSRZLLG0 | Report RSRZLLG0 Started | RSRZLLG0 |
| RSRZLLG0_ACTUAL | Report RSRZLLG0_ACTUAL Started | RSRZLLG0_ACTUAL |
| SAPMHTTP | Logon successful (type=H, method=P ) | H |
| SAPLSMTR_NAVIGATION | Transaction MM41 Started | MM41 |
| SAPMHTTP | Logon successful (type=H, method=P ) | H |
| SAPMHTTP | Logon successful (type=H, method=P ) | H |
| SAPMHTTP | Logon successful (type=H, method=P ) | H |
| SAPLSMTR_NAVIGATION | Transaction SE38 Started | SE38 |
| RSABAPPROGRAM | Report RSABAPPROGRAM Started | RSABAPPROGRAM |
| SAPMHTTP | Logon successful (type=H, method=P ) | H |
| SAPMHTTP | Logon successful (type=H, method=P ) | H |
| SAPMSYST | Logon successful (type=A, method=P ) | A |
| SAPLSMTR_NAVIGATION | Transaction ME21N Started | ME21N |
| RM_MEPO_GUI | Report RM_MEPO_GUI Started | RM_MEPO_GUI |
| RSAU_SELECT_EVENTS | Report RSAU_SELECT_EVENTS Started | RSAU_SELECT_EVE |
| RSRZLLG0 | Report RSRZLLG0 Started | RSRZLLG0 |

*9.2. Data Pre-Processing*

A substantial amount of the datasets obtained included text. Therefore, text pre-processing is mandatory for the analytics engine to make sense of the data and to be able to properly diagnose and suggest a solution. Below is an example of how we pre-processed text:

- *Tokenization*: in text analytics, a *document* is one;
- Piece of text, no matter how large or small. A document is composed of individual *tokens*. For now, think of a token or term as just a *word*. A collection of documents is called a *corpus*;
- *Filter tokens*: tokens that are less than 3 letters are filtered out;
- *Filter stopwords*: stopwords need to be removed. A stopword is a very common word in English. The words *the, and, of,* and *on* are considered stopwords in English so they are typically removed;
- *Normalization*: the case has been normalized: every term is in lowercase. This is so that words such as SALES and Sales are counted as the same thing;
- *Stemming*: word suffixes are removed, so that verbs such as announces, announced, and announcing are all reduced to the term *announc*. Similarly, stemming transforms noun plurals to the singular forms, which is why *directors* in a text becomes *director* in the term list;
- *n-grams*: n-gram is a bag-of-words representation which treats every individual word as a term, discarding word order entirely. In some cases, word order is important. For example, we could include pairs of adjacent words so that if a document contained the sentence *"PO down"* where PO stands for Purchase Order, which means the user is unable to create a purchase order as service is down. Generally speaking, n-grams are token sequences of length n.

### 9.3. Data Analytics

Many analytical techniques have been used such as association rules, TF-IDF, topic modeling, K-means clustering, ANN, and decision trees. In the next section, we present in a tabular format to show how we have used various analytics techniques to address which problem and the solution confirmation attained.

It is worth noting here that the analytics part is still ongoing to automate the analytics process as well as the discovery and application of a suggest solution.

### 9.4. Results Interpretation

In Table 2, we portray a partial list of the analytics we have accomplished so far. It should be noted that we continue to work to incorporate more problems and solutions. Also, we aim to fully automate the AAE engine phases and make it generalized enough to be integrated with more ESs.

**Table 2.** AAE Results.

| Problem Identification | Solution Building | Confirmation |
| --- | --- | --- |
| The ability to recognize which document is passed on to the AAE | Top-ranked tokens | Using text pre-processing techniques and top-ranked token, the AAE has been able to recognize and classify the document correctly (100% accuracy) |
| The ability to recognize general implementation problems | Year-to-support ticket correlation | Using text pre-processing techniques and the correlation coefficient, the AAE has been able to recognize that the implementation of the Human Resources (HR) module has been problematic |
| The ability to associate problems with solutions | Topic modeling | The use of topic models has enabled the AAE to associate certain problems and specific solutions with 80% accuracy. Via confirmation with case study organization officials, we reported 8 out of 10 cases were successfully grouping problems with the right solution within the same topic |
| Predicting the number of tickets a user will generate | Various prediction models | Using random forest, Neural Networks (NN), and Generalized Linear Model (GLM), the AAE has been able to predict tickets (avg. RMSE 2.3) |
| Classify the ticket to the appropriate module | Various prediction models | Using GLM, decision trees, and NN, the AAE has been accurately able to classify tickets to the appropriate module (avg. accuracy 56%) |

To explain how these results help to address the problem, we say that the AAE comes in two phases—as explained in Section 4—which are:

- Phase I: the AAE will analyze the environment. Here the results above helped to provide data insights to identify business blueprint documents and analyze them, as well as analyze support tickets and know which areas have more problems, linking tickets to modules.
- Phase II: the AAE will provide analytical capabilities related to failure prediction, solution confirmation, and full integration. Here the results above helped to link problems to solutions as well as predicting tickets and failures.

Nevertheless, we are still working towards providing more analytical and confirmatory components to be added to the AAE.

## 10. Generalizability

Ideally, DSR ought to involve the possibility of generalizing. In reality, the AAE as the artifact resulting from our DSR is a collection of ingredients serving different purposes. Therefore, it is a sort of a solution that addresses a specified problem. Hence, they both could be generalized. Sein et al. (2011) suggested four levels of generalization and we discuss how they apply to the AAE artifact:

(1) *Problem-related generalizations*: the AAE is designed to address problem-related operationalization of ESs. However, the AAE could also pertain to similar operationalization situations without ESs. For instance, principles could also apply to operationalizing embedded systems.

(2) *Solution-related generalizations*: the components of the AAE could be used as part of other solutions, such as the data acquisition component. The same goes for the analytics component.

(3) *Design principles*: the knowledge gained via design and implementation of the AAE could serve as design principles, for future researchers and practitioners to use. Below is the list of design principles which we have elucidated during the journey of the AAE design:

   - *Make ESs open*: it is important that the implementation of ESs takes into account openness, i.e., these systems need to be implemented, keeping in mind that other (analytical) systems will interact with these systems frequently.
   - *Get ESs ready for analytics*: the readiness of ESs for analytics is a key success factor, i.e., implementation and configuration of the systems needs to be documented and updated, since these documents will serve as environmental essentials for the analytics phase.
   - *The importance of post-live environment*: the post-live environments needs to be planned ahead, i.e., the role of the external vendors needs to be well-defined and documented. Otherwise, if the role is not well-defined and communication is not documented it is going to be impossible to automate it via the analytics engine.
   - *How to create support tickets*: creation of support tickets could represent the biggest hurdle towards the automation of solving tickets via the analytics engine. We have found that the more streamlined the ticket is, the easier it is to find a solution. By contrast, the more storytelling it contains, the greater the difficulty involved solving it. We have suggested to the retailer whom we obtained the data from to give training on how to create support tickets and to put as many options as possible and reduce free-text space.
   - *The necessity to properly configure application log files*: the application log is the place where many parameters are stored in time windows. Therefore, configuring it properly will help the analytics engine to read all necessary variables and hence make the right predictions about failures. Additionally, there must be enough retention policy for logs. For instance, less than a month is not going to help in predictions.
   - *It is all about text*: most of the sources come in the form of text. Accordingly, it is rather important to equip the analytics engine with enough text analytics capabilities—at the least,

text pre-processing, n-grams, TD-IDF, and topic modeling. That will enable the analytics engine to understand the business blueprint, support tickets, and emails and forums.

(4) *Feedback to design theory*: This type of generalizability constructs the cumulative efforts described above. It determines the range of generalizability for the design principles. This may include the creation of design theories or the possible refinement of contributing theories. That said, a complete design theory may not emerge from a single DSR endeavor.

## 11. Conclusions

Incorporating the AAE results formulates a balanced process, tackling the complications of traditional support activities through the ability of building an analytical discourse around given ESs spanning both the functional as well as technical underpinnings of the systems at hand. Through a plethora of currently proven advanced analytical methods applied to the inputs, such as business blueprints, support tickets, forums, emails, and system documentation, problems arising from traditional support process could be avoided. By analyzing these semi-structured sources, the AAE could come to an analytical discourse. This analytical discourse works on analyzing the historical and current health stance of the system to either propose relevant solutions or completely apply problem fixes.

This research work has discussed a suggested framework that works on addressing the ES post-implementation challenges by making use of a modular AAE. The six-phased DSR model has been adopted to help pave the research path. This is to be achieved through the design and development of the DSR artifact. Through several iterations, design and framework build of the artifact has been formulated to realize a modular advanced analytics agent. The agent works by integrating with any ES to analyze multiple data sources leading to resolutions safeguarding a systems' post-implementation activities. Automatic problem identification and solution confirmation could be done by the AAE through interaction with the ES. System modularity also explains how the system would be able to be hosted in the cloud. That said, future scholarly publications should illustrate and discuss observed findings from the design, development, demonstration, and evaluation phases by assessing use-case scenarios over several real-life cases, while also adopting the action design research methods to ensure organizational relevance [68]. Given the business impact, a next-generation breed of ESs that works on catalyzing business performance with an impact towards higher availability would pose a transformational stance towards maintenance and usage role personnel either in house (internal) or third party (external).

## 12. Limitations and Future Work

Regarding limitations, this research is still ongoing with a near-future focus on the issues of data management and fully automated data acquisition and pre-processing, which are also evolving.

Among the limitations, we have not studied the problems of end users interacting with the AAE. Added to that, we have also not focused on the challenges of implementing the AAE itself, in terms of being a socio-technical element introduced into the organizations seeking to solve their ES post-implementation challenges.

Regarding future work, given the impact of the AAE, the way organizations and vendors are going to engage will be transformed given the addition of AI acumen to the equation. This would correspondingly lead to transformational views not only on how internal and support personnel are to be assigned, but also on the supporting state-of-the-art technologies being part of the equation. With an outlook to the near future, it will significantly boost the levels of support communication to provide a standardized structure between stakeholders supporting the maintenance lifecycle. Correspondingly, it will require an update to the current academic ERP implementation lifecycle model. On a longer outlook, combining the explained framework with the bigger scopes covered by a wider in-scope breed of techniques is also required.

It was earlier explained that our designed AAE seeks to be artificially intelligent. In turn, the AI trait seeks to be enforced by extending the analysis scope with a plethora of methods (audio analysis, social network analysis, simulation modeling, optimization, etc.) to reach saturation. Henceforth, this opens wider research doors for studying how business process integration would be feasible, especially in the light of peer technologies (IoT, blockchain, robotics, etc.). That said, future research work will both encompass studying the application of state-of-the-art technologies as well as business related-organizational impacts.

It should be noted that future research also includes testing the AAE among end users in various organizational contexts. We strongly believe that will also create more room for new design principles being added to the ones already identified earlier on.

Longitudinal case studies are in the making, with organization adoption of the AAE reinforcing analytical capabilities.

**Author Contributions:** By referencing the MDPI Contributor Roles Taxonomy (CRedit), the following contributions had been acted upon by the respective authors. These contributions are: conceptualization, A.E. and H.E.H.; methodology, H.E.H.; software, A.E.; validation, A.E.; formal analysis, A.E. and H.E.H.; investigation, H.E.H.; resources, H.E.H.; data curation, A.E.; writing—original draft preparation, H.E.H.; writing—review and editing, A.E.; visualization, A.E & H.E.H.; supervision, A.E.; project administration, H.E.H.; funding acquisition, H.E.H.

**Funding:** This research received no external funding. H.E.H. did own the build-up and provisioning of the explained architecture in Figure 5.

**Acknowledgments:** The research team would like to acknowledge Rapidminer for providing the necessary Rapidminer educational licenses. These licenses had been leveraged to build the research artifact contributing to the evaluation of our scholar work. The research team would also lie to acknowledge the Egyptian retailer who accepted to provide data and validation to the experimental results.

**Conflicts of Interest:** The authors declare no conflict of interest.

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
