# Peer review of "Augmenting Advanced Analytics into Enterprise Systems: A Focus on Post-Implementation Activities"

_systems, doi:10.3390/systems7020031_

Reviewer 1 Report

The paper presents an interesting approach to the augmentation of advanced analytics into Enterprise Systems with a focus on post-implementation activities.

In the current form the paper has a relevant drawback, the lack of experimental results that can prove the value of the proposed principles and models.

The authors should improve the presentation of the following:

The state of the art research in the domain should be extended

the proposed models and architectures should be better correlated with existing models from literature 

The value added, benefits and impact of the research is not clearly presented. This aspects need to be further discussed. 

A validation method for the proposed model and architecture is not clearly described. 

Although the architecture is discussed, the presentation should be better structured (possibly by following the proposed components). Details regarding the implementation should be extended.

The presented concepts are not clearly analysed in comparison with other results from other researchers. As the work presented is work in progress the proposed concepts should be better argumented.

The presentation of the results in the section AAE evaluation is hard to follow and needs to be correlated with the presented model and architecture as well as compared to other existing implementations from literature.

Author Response

Reply to Reviewer 1

Dear Reviewer,

Thank you so much for taking the time to review our paper. We also appreciate your constructive comments to enhance the paper. In light of the comments which we have received from you, we have managed to enhance our paper. Please see below how we have addressed your comments in the new version of the paper:

·       In the current form the paper has a relevant drawback, the lack of experimental results that can prove the value of the proposed principles and models.

o   >> thank you for the note, we have presented our findings of experimental results in section 9 “AAE Evaluations”. On which, we have presented the data which we have sourced from a case study organization, that was also followed by data pre-processing activities as well as analytics. We, additionally, provided results interpretation in section 9.4 we certainly believe more analytics could have been done later in future research where, perhaps, we obtain more case organizations/datasets.

The authors should improve the presentation of the following:

·       The state of the art research in the domain should be extended

o   thank you for the note, in response to that note we have updated the SoTA research. Please see exemplar reference added:

21.         Shaul, L.; Tauber, D. Critical Success Factors in Enterprise Resource Planning Implementation: Review of the Last Decade. ACM Comput. Surv. 2013, 45, 10–21.

22.         Soja, P.; Paliwoda-Pekosz, G. Impediments to enterprise system implementation over the system lifecycle: Contrasting transition and developed economies. Electron. J. Inf. Syst. Dev. Ctries. 2013, 57, 1–13.

35.         van Beijsterveld, J.A.A.; van Groenendaal, W.J.H. Solving misfits in ERP implementations by SMEs. Inf. Syst. J. 2016, 26, 369–393.

42.         Chadhar, M.; Daneshgar, F. Organizational Learning and ERP Post-implementation Phase: A Situated Learning Perspective. J. Inf. Technol. 2018, 19, 138–156.

43.         Tian, F.; Xu, S.X. How Do Enterprise Resource Planning Systems Affect Firm Risk ? Post-Implementation Impact. MIS Q. 2015, 39, 39–60.

& Sections updated 2 & 4.

·       the proposed models and architectures should be better correlated with existing models from literature 

o   thanks for the note, however, in section 8.4 and 8.5, respectively, we provided details for the Technical architecture and the analytical techniques. To the best of our (limited)  knowledge this is the first study into the subject of integrating advanced analytics into enterprise systems. However, we are open to suggestions of similar papers. For instance, when we google (Scholar) AI and Enterprise Systems, we do not have any relevant papers, all are old and/or irrelevant.

·       The value added, benefits and impact of the research is not clearly presented. This aspects need to be further discussed. 

o   thanks for the not, we have updated sec. 6’s second paragraph to become in line with your comment

·       A validation method for the proposed model and architecture is not clearly described. 

o   thanks for the note, however, in section 9.4 we have provided results interpretation where modes used were tested for their accuracy, please revisit table 2. We have followed, to the best of our knowledge, evaluation as per the DSR guidelines in this highly recognized reference in the filed: Venable, J.; Pries-Heje, J.; Baskerville, R. A Comprehensive Framework for Evaluation in Design Science Research. LNCS 2012, 7286, 423–438

·       Although the architecture is discussed, the presentation should be better structured (possibly by following the proposed components). Details regarding the implementation should be extended.

o   thanks for the note, however, in section 8.4 and 8.5, respectively, we provided details for the Technical architecture and the analytical techniques. Unfortunately, we found it difficult to go to further details as believe size of the paper will substantially increase and may be the readers will lose focus

·       The presented concepts are not clearly analyzed in comparison with other results from other researchers. As the work presented is work in progress the proposed concepts should be better argumented

o   Thanks for the note, the research we have conducted, to the best of our (limited) knowledge is novel. The figure, Fig. 3, presenting the AAE is considered unique and we believe it advances the state-of-the-art further whereby advanced analytics is augmented into enterprise systems, compared to being used to analyze big datasets. In section 8, we have provided explanation of how we build the artifact and we believe this is our argument which we hope is sufficient giving the scarcity of similar studies 

·       The presentation of the results in the section AAE evaluation is hard to follow and needs to be correlated with the presented model and architecture as well as compared to other existing implementations from literature

o   Thanks for the note, but again we have presented our findings of experimental results in section 9 “AAE Evaluations”. On which, we have presented the data which we have sourced from a case study organization, that was also followed by data pre-processing activities as well as analytics. We, additionally, provided results interpretation in section 9.4 we certainly believe more analytics could have been done later in future research where, perhaps, we obtain more case organizations/datasets. About the note on literature, to the best of our (limited)  knowledge this is the first study into the subject of integrating advanced analytics into enterprise systems. However, we are open to suggestions of similar papers. For instance, when we google (Scholar) AI and Enterprise Systems, we do not have any relevant papers, all are old and/or irrelevant.

Reviewer 2 Report

Review of the paper:

Augmenting Advanced Analytics into Enterprise Systems: A focus on Post-implementation Activities

I read this paper with great interest. The paper has novelty and shows creativity, it is also very well written.  

It is a recognized problem from the literature that organizations struggle to optimize their ERP systems. Consequently, unsuccessful implementations occur – and the expected ROI is not achieved.

The authors of this paper seek to contribute to the solution of this problem and suggest advanced analytics to make enterprise system become more self-moderated to avoid problems in the post-implementation stage.

The paper provides a good and thorough overview of different aspects of the ERP lifecycle with a particular focus on the post-implementation stage. The authors highlight that all the stages in the ERP lifecycle are related – and failure in the pre-stage, make consequences for the post-stage.

The authors elaborate on ROI, performance dip, workarounds, etc. which are all important concepts for the post-stage.  The problems with relying too much on external consultants are also described.

The AAE artifact is technically very well explained with many details and figures.

Some suggestions that might help the authors in improving their paper:

It is not completely clear how the analytical capabilities of the AAE differ from a business intelligence system?  Could you elaborate a bit more on the uniqueness of the AAE and why it differs from other intelligent module-based systems (that can be add-ons to ERP systems)?  

One of the contributions of the paper is the set of design principles which add to our understanding of ES/AAE as a class of intelligent information systems. The design principles are a very interesting part of the paper – and should be highlighted and elaborated more on. The AAE (including the framework and the ES) is the artifact and together with the principles, it is part of an ensemble. Based on your material – I am sure there are more principles that you can add and elaborate more on? Section 10 is important as an argument for your contribution. I think you can go back to the design research of Hevner et al., and the ADR of Sein et al. – to make your contribution more specific. 

All the technical details are very well described; however, the paper is quite long. Maybe you can reduce some of the technical descriptions to make more space for your contribution to design research?

I disagree to take a solely technical view on the AAE. I think it has a major social part as well. The unstructured text-based sources which are part of the architecture, are socio-technical of nature, and you also have manual interpretations of the outputs of the system. The outputs of the system should be used by the organizational members to improve and optimize the ERP system and make them more independent of external support. The social learning process that the system creates, is important.

Future work; it is important that the AAE is tested among end users – and you probably need several iterations to update the design of the prototype to get it optimized. I think this design process should be elaborated on under your future work section. This is also feedback to design theory – that you are in the first stage. Also, the principles might need updates during future cycles (you mention this briefly in 9.3).

Limitations; I think the limitations of the study also deserve some reflections. The technical link between the ERP system and the AAE – is well explained – but how can the AAE identify problems with end users not understanding the system or working around it? One reflection here that can be included (limitation) is a potential “dark side” of the AAE – do we introduce another challenging system for the users? Can the implementation of the AAE itself cause challenges, not working as expected, or not being used according to its intentions? I think the idea behind the AAE is very interesting, but I think it is important to reflect on the challenges involved when implementing it.

Minors:

Figures and tables – not all of them have a table and figure text – please check. Consider if some of those can be moved to an appendix. E.g. those under 9.1 data acquisitions.

114 should be corrected to “dimensions”

341 should be corrected to “steps to be used to”

347 please correct to “an actual ERP case study”

422 please correct to” the problem is unprecedented

I wish the authors good luck with improving their interesting research.

Author Response

Reply to Reviewer 2

Dear Reviewer,

Thank you so much for taking the time to review our paper. We also appreciate your constructive comments to enhance the paper. In light of the comments which we have received from you, we have managed to enhance our paper. Please see below how we have addressed your comments in the new version of the paper:

·       It is not completely clear how the analytical capabilities of the AAE differ from a business intelligence system?  Could you elaborate a bit more on the uniqueness of the AAE and why it differs from other intelligent module-based systems (that can be add-ons to ERP systems)?

o   this is a very good point. We believe that the techniques which as have described and used in section 8, are also used and adopted in a business intelligence system. If we define business intelligence as an umbrella term which supports the decision making process and quality in originations, our AAE is not a business intelligence system as it does not make decisions which managers or decision makers do, but it instead uses the same techniques and algorithms used by the BI system. Additionally, our AAE adds a level of automation in order to augment the process of safeguarding post-implementation stage versus productivity decline

·       One of the contributions of the paper is the set of design principles which add to our understanding of ES/AAE as a class of intelligent information systems. The design principles are a very interesting part of the paper – and should be highlighted and elaborated more on. The AAE (including the framework and the ES) is the artifact and together with the principles, it is part of an ensemble. Based on your material – I am sure there are more principles that you can add and elaborate more on? Section 10 is important as an argument for your contribution. I think you can go back to the design research of Hevner et al., and the ADR of Sein et al. – to make your contribution more specific. 

o    we agree with the comment, but those are the DS’s which we thing apply to our research. We have put a limitation on the DS which could be generated in future studies e.g., end-user interaction with the AAE

·       All the technical details are very well described; however, the paper is quite long. Maybe you can reduce some of the technical descriptions to make more space for your contribution to design research?

o   thank you for the idea. We have already discussed design research in section 10; where we also introduced the design principles. The justification for the too much technical content is attributable to the nature of the AAE. We wanted to be very elaborate so as to encourage both academics and practitioners to look at the details of the engine in order to apply it themselves

·       I disagree to take a solely technical view on the AAE. I think it has a major social part as well. The unstructured text-based sources which are part of the architecture, are socio-technical of nature, and you also have manual interpretations of the outputs of the system. The outputs of the system should be used by the organizational members to improve and optimize the ERP system and make them more independent of external support. The social learning process that the system creates, is important.

o   we agree with the socio-technical note and added it as a limitation in the beginning of section 12

·       Future work; it is important that the AAE is tested among end users – and you probably need several iterations to update the design of the prototype to get it optimized. I think this design process should be elaborated on under your future work section. This is also feedback to design theory – that you are in the first stage. Also, the principles might need updates during future cycles.

o   thanks for the note, fixed in section 12

·       Limitations; I think the limitations of the study also deserve some reflections. The technical link between the ERP system and the AAE – is well explained – but how can the AAE identify problems with end users not understanding the system or working around it? One reflection here that can be included (limitation) is a potential “dark side” of the AAE – do we introduce another challenging system for the users? Can the implementation of the AAE itself cause challenges, not working as expected, or not being used according to its intentions? I think the idea behind the AAE is very interesting, but I think it is important to reflect on the challenges involved when implementing it.

o   thanks for the note, fixed in the beginning of section 12

Minors:

·       Figures and tables – not all of them have a table and figure text – please check. Consider if some of those can be moved to an appendix. E.g. those under 9.1 data acquisitions.

o   the suggestion is sound, however, we have opted to keep them under that section since it will enhance readership and does not require the reader to go back and forth eve now and then to follow-up with the text flow

·       114 should be corrected to “dimensions”

o    fixed

·       341 should be corrected to “steps to be used to”

o    fixed

·       347 please correct to “an actual ERP case study”

o    fixed

·       422 please correct to” the problem is unprecedented

o    fixed

Round  2

Reviewer 1 Report

The paper still has a relevant drawback, the experimental results are insufficient to prove the value of the proposed principles and models.

The experimental results focus on the data acquisition process. The data analytics process, which may be of interest for the reader, is mentioned, but not detailed. The data analytics should be further detailed.

The results interpretation should be extended and further detailed with specific cases for each conclusion. For example, the authors state:

“The use of topic models has enabled the AAE to accurately, 80%, associate certain problems and specific solutions “

What topic models were used in a specific case ?, How did the authors come to the result of 80% What problems where associated which what solutions ?

Author Response

Reply to Reviewer 1

Dear Reviewer,

Thank you so much for taking the time to review our paper. Please see below our answers to your comments:

-         The paper still has a relevant drawback, the experimental results are insufficient to prove the value of the proposed principles and models.

¾    As a matter of fact we have received this comment in previous round. We certainly do not want to repeat ourselves. But in brief, we have presented our findings of experimental results in section 9 “AAE Evaluations”. On which, we have presented the data which we have sourced from a case study organization, that was also followed by data pre-processing activities as well as analytics. We, additionally, provided results interpretation in section 9.4 we certainly believe more analytics could have been done later in future research where, perhaps, we obtain more case organizations/datasets.

¾    Added to above, we have thought about working with synthetic data since at the moment we do not have other dataset but the one presented in the paper, but then opted for not using it. as according to literature, it is not advisable to use where the outcome represents knowledge which would be used organizationally. Please see: D. Libes, D. Lechevalier, and S. Jain, “Issues in synthetic data generation for advanced manufacturing,” Proc. - 2017 IEEE Int. Conf. Big Data, Big Data 2017, vol. 2018-January, no. December 2017, pp. 1746–1754, 2018.

¾    Lastly, we are in the mid of a process by which we will acquire more real dataset and apply more analytics on our AAE framework.

-         The experimental results focus on the data acquisition process. The data analytics process, which may be of interest for the reader, is mentioned, but not detailed. The data analytics should be further detailed.

¾    In our analysis, we have followed an adapted version of the CRISP-DM mythology for data mining (https://www.the-modeling-agency.com/crisp-dm.pdf). Accordingly to which, and to other studies, 80-85% of the time in analytics is spent on acquiring, understanding, and preprocessing the data. So, if our paper gives that impression to the reader I think we are on the right track according to industry and academia. Kindly see: Müller O, Junglas IA, Brocke JV, Debortoli S. Utilizing big data analytics for information systems research: challenges, promises and guidelines. Eur J Inf Syst (EJIS). 2016: 1–14. Also: Elragal, A. & Klischewski, R., (2017) “Theory-driven or Process-driven Prediction? Epistemological Challenges of Big Data Analytics”. Journal of Big Data, 4:19, DOI 10.1186/s40537-017-0079-2.

¾    Added to that, we wanted to give the reader substance on the complexity of analyzing textual datasets and how to get around them.

-         The results interpretation should be extended and further detailed with specific cases for each conclusion. For example, the authors state: “The use of topic models has enabled the AAE to accurately, 80%, associate certain problems and specific solutions”. What topic models were used in a specific case ?, How did the authors come to the result of 80% What problems where associated which what solutions?

¾    The results have been explained using various techniques which are relevant to the dataset in hand and the nature of problem e.g., text analytics using topic modeling, correlations, GLM, and random forest.

¾    The question about topic modeling: in our analysis, we have used the Latent Dirichlet Allocation (LDA) algorithm. In topic modeling, the assumption is that words that co-occur together in similar contexts tend to have similar meanings. Hence, sets of highly co-occurring words could therefore be interpreted as topics  and used to cluster documents into topical classes. The LDA is a well-known and highly cited topic modeling algorithm that is able to discover topics running through a corpus and to annotate individual documents with topic labels. As an unsupervised machine learning algorithm LDA is purely data-driven and inductively infers topics from given texts.

¾    About the 80%, the topics have been validated by the case study organization whereby in 8 out of 10 cases, topics detected were able to associate the problem with the solution.

¾    We have added a line to explain where the 80% came from in Table 2. [via confirmation with case study organization officials, reporting 8 out of 10 cases were successfully grouping problems with the right solution within the same topic]

¾    Lastly, we have indeed thought about adding more about analytics. However, we decided to do it on a different other paper as that other paper will be technique and/or algorithmic oriented. On the other hand, our current paper is focusing on building the IT artifact. We have also received another comment on the volume of the paper which makes it difficult to add more analytical details.
